# Evaluating the Relationship between Well-Being and Living with a Dog for People with Chronic Low Back Pain: A Feasibility Study

**DOI:** 10.3390/ijerph16081472

**Published:** 2019-04-25

**Authors:** Eloise C.J. Carr, Jean E. Wallace, Rianne Pater, Douglas P. Gross

**Affiliations:** 1Faculty of Nursing, Professional Faculties Building, 2500 University Drive NW, Calgary, AB T2N 1N4, Canada; Rianne.pater@ucalgary.ca; 2Department of Sociology, Faculty of Arts, University of Calgary, 2500 University Drive NW, Calgary, AB T2N 1N4, Canada; jwallace@ucalgary.ca; 3Department of Physical Therapy, Rehabilitation Medicine, University of Alberta, 2-50 Corbett Hall, 8205 114St., Edmonton, AB T6G 2G4, Canada; dgross@ualberta.ca

**Keywords:** chronic pain, well-being, depression, physical activity, dog ownership

## Abstract

Chronic low back pain is a significant societal and personal burden that negatively impacts quality of life. Dog ownership has been associated with health benefits. This study evaluated the feasibility of surveying people with chronic low back pain to assess the relationship between dog ownership and well-being. A mail-out survey was sent to 210 adult patients with chronic low back pain. Measures of quality of life, pain, physical activity, emotional health, social ties and dog ownership were included. Feasibility was assessed by examining survey response rate, responses to established and newly developed measures, and the potential relationships between dog ownership and a number of key well-being variables in this patient population. There were 56 completed surveys returned (*n* = 36 non-dog owners and *n* = 20 dog owners). Established, adapted and newly developed scales revealed promising results. Dog owners reported fewer depression and anxiety symptoms, and more social ties than non-dog owners. Living with a dog may be associated with improved well-being for people with chronic pain. The findings from this feasibility study will inform a general population survey, to be conducted with a larger, more representative sample of people living with chronic pain.

## 1. Introduction

Chronic pain is one of the most important current and future causes of morbidity and disability across the world [1,2]. In the United States an estimated 20.4% of U.S. adults have chronic pain and 8.0% of U.S. adults have debilitating chronic pain [3]. In Europe, chronic pain of moderate to severe intensity occurs in 19% of adult Europeans [4]. One in five Canadians experience chronic pain, which costs more than cancer, heart disease, and HIV combined [5], with costs related to job loss and sick days estimated to be $37 billion per year [6]. Chronic pain is defined as pain that lasts longer than three months and is associated with significant distress and disability, including reduced physical activities and social ties [7].

Back and neck pain are the leading global reasons for disability [2]. In the United States, low back pain (LBP) is the most common type of chronic pain and the leading cause of disability in people under 45 years of age [8]. In Canada, the lifetime prevalence of LBP has been estimated as 84%, with 62% of respondents reporting LBP in the last year [9]. As already noted, the healthcare and personal costs can be enormous. People living with chronic LBP often report not only living with pain and a loss of physical function, but also a reduced social life, poor psychological well-being [10] and poor sleep that is known to reduce pain thresholds [11]. The medical, social and economic costs of chronic LBP are so large that it has been regarded as an epidemic [12]. A recent series of articles in the *Lancet* issued a ‘Call for Action’ to change how back pain is cared for [13]. A key recommendation is to move away from a biomedical model of care towards more “person-centered care focusing on self-management and healthy lifestyles as a means of restoring and maintaining function and optimizing participation.” Novel and cost-effective strategies were recommended; especially those that can be undertaken within community settings and that are easily translated to low- and middle-income countries.

The connection between human and animal health is garnering significant attention in both human and veterinary medicine, with a focus on the concept of the Human-Animal Bond (HAB), and how it relates to healthcare [14]. The unique and enduring bond between dogs and humans has been studied through the lens of different disciplines and has garnered significant research attention. The attachment bonds that form between dogs and humans have been compared to those between an infant and its mother [15]. A recent review considered research on the attachment bonds between humans and dogs, tools to measure the bond, and the effects of different bonds on the dog-human dyad as a whole [16]. They concluded that positive attitudes and affiliative interactions appeared to contribute to the enhanced well-being of both species, as reflected in resultant physiological changes. It has been suggested that neuropeptide oxytocin is the mediator for the psychosocial, neuroendocrine and biological effects of the human-animal interaction [17]. The HAB continues to be explored and in particular the influence it has on human health.

Owning a companion dog provides several benefits to owners’ well-being. We use the term well-being in the broadest sense to encompass physical, mental, and social domains of health to encourage a more holistic approach to disease prevention, disease management and health promotion [18]. Specifically, dog ownership has been associated with: improved physical activity [19,20,21,22], better mental well-being [23,24,25], greater connectedness to the community or ‘social capital’ [26,27] and reduced medical costs [28]. The benefits are so positive that dog-walking has been encouraged as a population outreach activity for health promotion and disease prevention [19]. A companion animal (e.g. cats, dogs) generally includes any animal that shares its life with a human care-giver [29]. However, we are particularly interested in people who own and live with a dog as they are more researched than other companion animals, more likely to be involved in physical activities outside the home, and are often identified as a member of the family [30]. 

We conducted qualitative interviews with people who had chronic pain and lived with a dog and our findings suggest there were physical, emotional and social benefits that improved their well-being [31]. We are particularly interested in the potential benefits of owning a dog, where there is a close bond to the dog, and the role this may play in the self-management of chronic LBP.

The aim of this study was to evaluate the feasibility of surveying people with chronic LBP to empirically assess the relationship between dog ownership and well-being for people with chronic LBP. Feasibility studies and pre-testing research instruments are an essential element in designing a successful, large-scale study. They can be helpful in: determining if the sampling frame and recruitment strategy are effective; developing and testing the adequacy of measures; and offering preliminary data about the relevance of the research question and its worthiness of future study [32,33]. As a result, the primary objectives of this study were to ensure that: (1) the recruitment strategy was effective for this population [34]; (2) the data collection instrument (paper questionnaire) was appropriate and survey response burden was not an issue; and (3) the newly developed and key established measures were valid and reliable for this population. The secondary objective was to explore whether a relationship exists between dog ownership and well-being (physical, mental and social domains) for people with chronic LBP.

## 2. Materials and Methods

### 2.1. Study Design, Setting and Sample 

The study used a survey design. Patients referred to a community multidisciplinary chronic pain program in western Canada were invited to participate if they had current addresses in the regional patient demographics system and met study inclusion criteria. The chronic pain program offers patients access to an interdisciplinary team comprising medical specialists, a clinical psychologist, nurses, physical therapists, occupational therapists and a case manager. Individual and groups sessions are offered on an out-patient basis, usually over a 12-month period. The program is tailored to meet individual needs and aims to improve function and quality of life. Inclusion criteria for this study were: participants must be ≥18 years, have a diagnosis of chronic LBP present for ≥ 6 months, and a pain score of ≥4 on a Numerical Rating Scale (NRS, 0–10) at the time of invitation to the study. Potential participants were identified from the clinic database using the diagnostic code for low back pain (ICD10 code M54.5).

The average sample size for detecting the occurrence of a problem in a feasibility or pilot study has been determined to be 30 participants per group [35] and a minimum of 30 participants is reasonable as a starting point for pre-tests of questionnaires [36]. Response rates to surveys can be variable but population surveys of chronic pain patients have achieved responses between 52% [37] and 62% [38]. We, therefore, estimated that a sample of 210 patients was adequate for this feasibility study. A data analyst at the pain program wrote a script to randomly identify patients at different stages of their chronic LBP journey including: those on a waitlist to see a physician at the program (n = 52); those on a waitlist to see a physician at the program and have participated in group sessions (n = 53); those actively being seen by a physician (*n* = 53); and those discharged after completing the program (*n* = 52). Surveys were then sent to these four groups of patients.

### 2.2. Survey Development

In developing the survey, we were interested in ensuring our measures reflected the multidimensional experience of chronic LBP related to pain and physical function, emotional health, and social interaction [10]. These broad areas provided the framework to organize the measures in the survey (refer to the survey in the Appendix A). Measures were chosen for their relevance to the research questions, desirable psychometric properties, and suitability for this target population (see below). Next, we incorporated a number of additional measures that reflected key constructs reported as important from interviews carried out in the first phase of the study [31]. For example, these included how living with a dog provided meaning and purpose to life, routine, companionship, and distraction from chronic pain. Some of these measures already existed in the literature [23,39,40,41] some were adapted from existing measures [42,43], and others were developed for this study based on the participants’ interview responses in the first phase of the study [31]. Our measurement assessment in this study focuses on these newly developed dog ownership measures. Once we had identified potential measures, they were grouped into four key areas: 1) Well-being and quality of life; 2) physical activity and physical health; 3) emotional and mental health; and 4) social and community ties. In addition, we devoted one section of the survey to a number of specific questions for those owning dogs that was largely influenced by the new concepts identified by the interview phase of the study. Participants were also asked a range of standard demographic questions (e.g. age, sex, family situation, education etc.).

#### 2.2.1. Well-Being and Quality of Life 

General health measures and two pain assessment scales were used to assess participants’ well-being and quality of life. Three items from the US National Survey of Families and Household were included that asked participants to rate their overall health, physical health, and mental health, compared to other people their age [44]. These items used a Likert response scale from 1 (poor) to 5 (excellent) health. These single-item, self-report measures have been shown to be highly correlated with more objective assessments made by physicians and various measures of morbidity [45]. Second, we included the Healthy Days Measure from the Center for Disease Control and Prevention [46]. This measure includes three items that ask participants to identify how many days, in the past 30 days, that their physical and mental health was poor, and whether it prevented them from doing their usual activities, self-care, work, or recreation.

Two pain measures were included that assess pain severity and physical functioning with LBP. Pain severity was measured using the pain numeric rating scale (NRS). The pain NRS is simple to understand, demonstrates good reliability, and is widely used in clinical and research settings [47,48]. The Oswestry Disability Index v2.0 (ODI) is seen as the gold standard for measuring LBP disability [49] with excellent reliability and validity [50]. It is a 10-item scale that measures LBP intensity and the impact of LBP on a range of functions including personal care, lifting, walking, sleeping, and social life, among other functions.

#### 2.2.2. Physical Activity and Physical Health

Physical activity was measured using the Godin Leisure Time Exercise Questionnaire (GLTEQ). Participants were asked to recall their average weekly physical activity during their free time over the past month [51]. This involved providing estimates for the number of times per week and the minutes each time that respondents participated in strenuous, moderate, and mild exercise. Leisure time walking was measured by Brown and Rhodes’ [52] adaptation of the Godin and Shepard [51] measure that includes estimates for the number of times per week and the average minutes per time of mild, moderate and strenuous walking with and without a dog.

#### 2.2.3. Emotional and Mental Health

Four measures were included to assess emotional and mental health. Emotional well-being was measured by the WHO-5 Index with five items assessing overall emotional well-being [53]. The scale has adequate validity both as a screening tool for depression and as an outcome measure in clinical trials, and has been applied successfully across a wide range of study fields [54]. It has also demonstrated robust psychometric properties and reliability in the evaluation of back pain treatments [55]. Depression and anxiety were measured by eight items from the Patient-Reported Outcomes Measurement Information System (PROMIS^®^) short form (v2.0) PROMIS anxiety SF4 and the PROMIS depression SF4 scale [42,56]. Two further measures were developed from our interviews with dog owners. One included five items adapted from the PROMIS Parents’ Meaning and Purpose short form (v1.0) scale measuring whether respondents’ life has meaning and purpose. The other included two items developed from the interviews that measures whether respondents’ life has meaning and structure. 

#### 2.2.4. Social and Community Ties

Social and community ties were measured by four scales. The Loneliness Scale and the Networks for Support Scale were used to measure social and community ties specifically in the dog ownership population [26,27,39]. The Companionship scale and Emotional Support scale (PROMIS, short form, v2.0) each contain 4 items and were used to measure social functioning [42,57]. 

#### 2.2.5. Dog Ownership Experiences 

For respondents who owned a dog, we asked additional questions related to the dog’s sex, age, size, primary reason for having the dog, dog health, and attachment. One of the most widely used instruments to measure the attachment pet owners have to their dogs is the Lexington Attachment to Pets Scale (LAPS) [58,59]. A series of measures were included to capture a wide array of possible benefits of dog ownership that included support [26], emotional benefits (a subscale of the Human-Animal Bond (HAB) Scale [40], companionship (a subscale of the Human-Animal Bond (HAB) Scale [40], social facilitation, physical benefits, quality of life [41], and sense of community [26]. These measures are based on scales previously used and tested in the literature on companion animals and their connections with and benefits for their owners [26,39,40,41]. We developed several additional measures based on the findings from our interviews and another qualitative study on the benefits of therapy dogs [31]. These items reflected a number of dog benefits such as dog as pain reliever, dog as stress reliever, and dog as unconditional support. The scale ‘dog gives meaning and purpose’ was adapted from the PROMIS (Short form v1.0) Parents’ Meaning and Purpose scale [42].

### 2.3. Patient Engagement

Engaging patients as partners in research can ultimately improve the quality of care and is particularly important for chronic health conditions [60,61]. Frameworks such as the Guidance for Reporting Involvement of Patients and Public (GRIPP2) checklist help ensure that reporting patient involvement in research is transparent [62]. However, the tool has been criticized for assessing the impact of patient engagement in research, rather than the quality of the reporting [63]. Another tool that is helpful to make the patient involvement in the research process more transparent, is the Patient Engagement in Research Description (PED), that identifies three over-arching categories of who, how, and when [64]. In this paper we use the PED framework to describe the involvement of patient members in the research process.

The Human Animal Pain Interactions (HAPI) research team has a core team of 15 members. The team includes five patient members, two members from Alberta Health Services, and the remaining 8 researchers are members of several disciplines including, nursing, sociology, veterinary and human medicine. All five of the patient members have chronic pain and two live with service dogs. In addition to bringing the patient perspective of living with chronic pain they have specific expertise. For example, three patient members have been active facilitators for a community patient support group for people with chronic pain who have completed a pain management program. Two patient members have been trained in the Patient and Community Engagement Research (PaCER) method, which is a peer-to-peer inductive research approach designed to create a robust shared patient voice and maximize patient engagement throughout the research process [65,66]. The ‘how’ of engaging patient members was achieved by including them as equal members of the research team. They were invited to attend meetings, included in email communications and invited to contribute on specific items that were important to the patient perspective. For example, prior to dissemination, the survey was reviewed by two patient members who lived with service dogs. As subject matter experts, they offered feedback on the questionnaire in terms of the clarity and relevance of measures, as well as the ease and timing of completing the questionnaire.

### 2.4. Data Collection

A mail-out package was sent to survey participants with a paper copy of the survey, a consent form and a pre-paid return envelope. The consent form included detailed information about the study. Participants were informed that we were conducting a study looking at people with chronic low back pain who may or may not own a dog and that we were interested to see if there were differences in health outcomes between people who owned a dog and those who do not. They were told how their names had been identified, that their responses were anonymous, and that we estimated it would take 15–20 minutes to complete the survey. There were no incentives or reimbursements for participation provided. A recent systematic review demonstrated that paper surveys had significantly higher response rates than electronic surveys and older adults are less likely to be regular internet users [67]. A review comparing paper and web-based surveys suggested that paper surveys had an average response rate of 56% compared to 33% for online surveys [68]. Reminder letters were sent as follow up three weeks later. Data collection took place between November 2017 and January 2018.

### 2.5. Data Analysis

In analyzing the survey data, five types of analyses were carried out. First, descriptive analyses of all of the items were analysed. Frequencies for all of the items were examined to see whether survey participants are using all of the response options on each item and the range of scores of each item. In addition, the completion rate on each item was noted. Then descriptive information was generated to determine characteristics of the sample in terms of their well-being/quality of life, physical activity and physical health, emotional/mental health, social/community ties, dog ownership experiences and attitudes and demographic variables. Second, scale development and exploratory factor analysis (EFA) was carried out to empirically examine the latent structure among the newly developed measures for the dog ownership part of the questionnaire. This included, for example, measures of the benefits of dog ownership such as dog as pain reliever, dog as stress reliever, and dog as unconditional support. Due to the small sample size for the dog owners who completed the survey (*N* = 20), only items expected to load on a single underlying construct were entered in order to see whether all of the items loaded on a single factor and did not cross load on another factor. For example, all five items expected to measure the dog benefit of “dog a pain reliever” were entered together but they were not entered with the other dog benefits items tapping “dog as stress reliever” or “dog as unconditional support”. Retention of items was based on factor loadings of ≥0.30 where loadings did not cross load on another factor. Items that failed to meet these criteria were then removed from the scale. Scale scores were computed by summing the respondents’ scores for each item and dividing by the number of items. Third, scale confirmation and principal confirmatory factor analysis (CFA) was used to confirm the factor structure of already existing measures, which were completed by all 56 survey participants. For example, the WHO Well-Being Index, and the PROMIS scales were subject to confirmatory factor analysis to determine whether they loaded on a single factor as expected. The same factor loading criteria outlined above were used to interpret the factor structures. Similarly, scale scores represent the mean scores of the relevant items. Fourth, the internal consistency, or reliability, of scale items was examined. To do so, Cronbach’s alphas were estimated for the items included in each scale. Fifth, mean difference tests were carried out by comparing the average scores of dog owners (*N* = 20) with non-dog owners (*N* = 36). In addition, zero-order correlations were also estimated to examine the relationships between pain severity, physical health and functioning, social/community ties and depression. 

### 2.6. Ethics

The study received ethical approval from the University of Calgary Conjoint Health Research Ethics Board (#REB17-0829).

## 3. Results

The results are organized around the objectives regarding the recruitment strategy, the appropriateness of the data collection instrument, issues related to the measures of key variables, and the relationship between dog ownership and well-being.

### 3.1. Recruitment and Response Rate

A total of 56 completed surveys were returned from a mailing out to 210 to people with chronic back pain who had attended a community multidisciplinary pain program in western Canada. Of the 210 surveys mailed out, 7 were returned to sender as the recipient was no longer residing at the address.

The process of recruitment worked well but the 27% response rate was less than anticipated. However, as shown in Table 1, comparing the demographic characteristics between the two groups, dog owners and non-owners, shows that they did not differ significantly by sex, age, education, available income, housing situation, urban/rural location, employment status or marital status. Whilst this is a lower than anticipated response rate we suggest that the overall length of the questionnaire did not overly deter participants from responding, despite having 174 items for dog-owners and 117 items for non-dog owners. As discussed below, certain sections however had lower completion rates within the survey. We were also surprised to find that many respondents had provided additional written comments, which are discussed in greater detail below.

### 3.2. Sample Characteristics

The 56 completed surveys included 36 non-dog owners and 20 dog owners. The participants included 21 men and 35 women. Most (67.3%) were married or in a common law relationship and 19.6% had a child under 18 years of age living at home (see Table 1). They ranged in age from 26 to 82 years (Mean = 56, SD = 13.08). In terms of employment, 48.2% were unable to work, 19.6% were retired, 17.9% were employed or self-employed and 14.3% were out of work or a homemaker. The majority had some post-secondary education (72.7%). Comparisons were made between the dog owners and non-dog owners on all demographic variables and none were significantly different at the 0.05 level

### 3.3. Completion Rate

The data collection instrument was a 10-page paper questionnaire and some participants may have been deterred by its length. In terms of analyzing the completion rates for specific sections, five of the six sections had only three or four missing responses on most items. There are two exceptions however. First, in the opening section of the survey on well-being, three items asked participants about the number of days they experienced poor physical health, poor mental health and were unable to do their usual activities over the last 30 days [46]. Missing data on these items ranged from seven to 15 responses. It is not clear if participants intentionally left the responses blank indicating zero unhealthy days or skipped the question. Second, there was significant missing data in the section measuring physical activity for most of the 34 items. Missing responses varied between three to 43 respondents not completing the items (results available from authors) and only 23% of participants fully completed this section. Participant fatigue could have been an issue because some items required too much detail and calculations (refer to Section Two of the survey in the Appendix A). This measure was included because it is popular in dog walking research where one of the primary goals is to assess the link between owning a dog and frequency and intensity of physical activity.

### 3.4. Measures

Our focus is the measurement properties of the newly developed measures tapping the benefits of dog ownership. As mentioned above, scale development and exploratory factor analysis (EFA) was carried out to empirically examine the latent structure among the newly developed measures for the dog ownership part of the questionnaire. Only items expected to load on a single underlying construct were entered in order to see whether all of the items loaded on a single factor and did not cross load on another factor. Table 2 shows the items that were retained for each of the dog benefit scales and all of these newly developed measures also indicate good reliabilities of 0.70 or higher. All pre-existing scales were found to represent a single underlying factor as expected. Upon completion of the factor analyses, reliability of the multiple item measures was assessed using Cronbach’s alpha (α). Values of 0.70 or higher indicate an acceptable level of internal consistency among the items [69]. All of the pre-existing scales had reliabilities of 0.80 or higher (results available from authors).

### 3.5. Exploration of the Relationships between Dog Ownership and Well-Being

We then evaluated the relationship between dog ownership and well-being to assess the feasibility of exploring this relationship further with a larger sample. We selected several key variables from the survey that have been linked to the benefits of dog ownership (i.e., physical and social benefits) that also are related to improved well-being [70]. The results in Table 3 show that despite dog owners and non-dog owners reporting similar pain severity scores, both groups report relatively high pain scores. It is also interesting to note that the two groups do not differ significantly in their physical functioning or physical health. Again, the magnitude of these scores suggest that our sample suffers from severe disability in performing daily living activities related to their chronic pain. Turning next to the social and community ties measures, we see that the two groups are similar in their degree of loneliness and emotional support.

The dog owners, however, report more social ties in terms of people offering companionship. Lastly, the depression scores differ significantly such that the dog owners reported fewer depression and anxiety symptoms over the last week before the survey than the non-dog owners. 

Table 3 also shows how each of the variables correlates with depression for the entire sample. For this we can see that, as expected, greater pain severity is correlated with greater depression. While the two groups do not differ significantly in physical functioning and health, overall physical function and pain walking were significantly related to feelings of depression. Also as expected, loneliness is positively related to depression and companionship and emotional support are negatively related.

### 3.6. Additional Comments on the Survey

At the end of the survey, respondents were invited to include any additional comments or thoughts that they believed could assist the researchers to better understand the experiences of living with chronic back pain. Of the 56 surveys returned, 16 survey respondents had included substantive feedback, defined as any additional comments or thoughts that were greater than two sentences. Seven (44%) of these comments were made by dog owners while the remaining nine (66%) were made by non-dog owners.

Of the sixteen comments, the majority focused on describing their pain and its management but several provided examples of how their dog helped them cope with their pain: *“My dog has very much improved my quality of life. He can be a challenge sometimes because he is a Pembroke Welsh Corgi and they can be stubborn. My life would be empty without him. He brings structure to my day. Touching him brings me calm. The softness of his fur seems to bring my feelings away from my pain and it’s like I can no longer feel it for a time. I could not imagine life without a dog.”* (P40).

However, both dog owners and non-dog owners mentioned negative aspects of dog ownership. For example, dog owners noted barriers to dog ownership in the context of chronic pain: *“my dog is much more difficult to care for during winter months due to the snow and ice… Summer is much easier, and I love spending time with my dog then”* (P23). Another participant had a similar experience: *“when it’s cold and sidewalks are covered in snow or ice I try to walk him at least once a day, but I am very afraid of falling and hurting myself.”* (P37). Non-dog owners referred to the financial constraints of owning a dog: *“I feel a dog would really benefit me. I have had dogs always but now can’t afford to buy a dog and the only way we would have a dog to benefit me would have to be a service dog. I love being around dogs and I think it would help me be more active.”* (P47). And: *“love to have a dog, but vet bills and food is a big problem when you are on disability and have low income.”*(P19)

## 4. Discussion

This study evaluated the feasibility of surveying people with chronic low back pain to assess the relationship between living with a pet and well-being. The findings from this feasibility study will inform a larger population survey and some changes in the protocol. We discuss these findings in relation to the recruitment processes; the survey instrument; the development of new and adapted variables, and the findings pertaining to the relationship between dog ownership and well-being.

The response rate of 27% was lower than expected as prior studies using surveys with chronic pain patients have achieved responses between 52% [67] and 62% [68]. One possible explanation for this lower than anticipated response could be related to the high level of pain and disability in the sample. Characteristics of patients attending chronic pain programs include poverty, financial stress, severe functional disability and past or current medical conditions [71]. We observed high scores for disability and the fact that questionnaires continued to be returned over a five-month period. Upon receiving the questionnaire, some may have been deterred by the length of the survey (10 pages) and/or some of the more complex questions as noted above (e.g. physical activity). To improve this response rate for future studies we would reduce the length of the survey. While our patient members of the research team reported that it took them 15–20 minutes to complete the survey, this still may have been too taxing for some patients. Due to budget constraints we only sent one reminder letter at three weeks, but in future studies we recommend sending a second reminder to optimise response rate [68]. As we only provided a paper copy of the survey it is difficult to comment whether providing an electronic version would have improved the response rate. We provided paper surveys as it has been reported that these provide higher response rates than electronic surveys as older adults are less likely to be regular internet users [67]. However, for future studies we recommend offering both modalities to optimise response rates. We were also surprised that more people responded who did not have a dog (*n* = 36) compared to those who did (*n* = 20). However, we had not taken into account that in Canada 32% of households own at least one dog [72]. These statistics are similar to those from other western countries. For example, in the UK 30% of households have at least one dog [73]. In the US at least half the people who live in the community between the ages of 50 and 74 years have a dog or a cat in their household, with 28% owning a dog [74]. In calculating sample size for future studies, we recommend accounting for the ratio of dog owners to non-dog owners and increasing the sample size accordingly. We were unable to locate any studies suggesting people with chronic pain are less or more likely to have a dog.

The preliminary analyses of the newly developed multiple-item scales included in this questionnaire yielded favorable results. Despite the small sample size, the established scales conformed to the expected factor structures, demonstrated high internal reliability and were related to depression consistent with the existing literature. These measures appear to be appropriate in assessing the well-being/quality of life, emotional/mental health, and social/community ties for this population. The low completion rate for the physical activity and walking items suggests these items may have been overly taxing for participants to fill out. As mentioned above, these items involved numerous estimates of the time and amount of time spent in activities of different exertion rates with and without a dog. For future surveys of people living with chronic pain, we recommend reducing this burden by using the 4-item short-form version of physical function measure from the PROMIS survey. It demonstrates good reliability and validity from chronic musculoskeletal pain and has been recommended for clinical research in low back pain [69]. Whilst the original purpose of PROMIS was for use in clinical trials it has been used across many different chronic disease populations. The PROMIS tools can be used for population surveys and across chronic conditions [75,76].

The newly developed measures of the benefits of dog ownership (i.e., dog as pain reliever, stress reliever, and unconditional support) also yielded promising results. Due to the small sample size of this feasibility study, rigorous validation tests could not be performed but should be the next step with a larger sample. Inclusion of these measures in future research could improve our understanding of the benefits of dog ownership for those with chronic pain.

We explored the relationship between dog-owners and non-dog owners and well-being and found differences between companionship and depression, but no differences in pain, physical function, loneliness or emotional support. Depression in people with chronic pain is common with up to 70% of patients experiencing depressive and anxiety disorders, but the underlying mechanisms are poorly understood [77]. It has been suggested that disability and social isolation, that often accompanies chronic pain, can lead to depression and anxiety, but also it can work in the opposite direction [78]. It makes sense that pain severity, companionship and emotional support would then be associated with depression. The relationship between dog ownership and depression is not consistent in the literature, with some studies finding dog owners have less depression [79] and others finding no difference [80]. These inconsistent findings are likely due to different measures, different populations and a lack of control for confounding variables. However, we suggest that the dog owners’ lower depression scores may be partly accounted for by less loneliness and their greater access to companionship that may be related to owning a dog. We conclude that further research in this area is feasible and may offer meaningful results to the existing literature on chronic pain and alternative approaches to chronic pain self-management. 

However, further exploration is warranted to understand if dog ownership might be a mediating factor for depression for people with chronic pain, and if so then how. Interestingly we found no difference between the two groups for physical functioning, but given the severity of their chronic pain and resultant disability it is not surprising that these people were not particularly active. 

Given that approximately 60% of people with back pain do not seek care within the health care system [9], future research using our questionnaire battery should likely involve the general population. Limiting the sample to clinical populations, as we did in this feasibility study, omits people who are coping with their pain and able to function independently in the community. It may be that dog ownership and its associated benefits are factors that are more beneficial self-management strategies for people suffering from less severe chronic pain than those who participated in our study. The open text comments on the survey provided further insights that should be considered for future studies, particularly around barriers to dog-ownership such as veterinary and food costs, and concerns about being able to exercise a dog. It could be that participants who had previously owned a dog but could not do so currently might hold different views to those who had never owned a dog. This warrants further exploration. A general population survey, conducted with representative sampling and computer-assisted telephone interviews or web-based surveys, would enable us to evaluate further the role of dog ownership in the self-management of chronic pain.

There were several limitations to our study. One of the major limitations was the modest response rate that may have contributed to bias in the sample. It will be important to build in ways to mitigate this in future studies, such as offering both paper and online survey options, and follow up reminders. To fully understand an individual’s responses to study variables the study design needs to be longitudinal and ideally include measures over a period prior to owning a dog [25,80]. In our study design it might be that people who own a dog are healthier than people who choose not to own a pet, a point that has been made by others [81]. Pet ownership has been associated with more affluent groups but our socio-economic demographic variables were similar for both dog owners and non-dog owners. A further consideration might be a study design allowing participants to be randomly assigned to groups and self-select pet ownership but this would be ethically and logistically challenging.

## 5. Conclusions

This study evaluated the feasibility of surveying people with chronic low back pain to assess the relationship between living with a pet and well-being. Whilst the survey items appear to have promise, the response rate and completion rates suggest the survey response burden was high, particularly for this population who are very debilitated by their pain. Capturing physical activity using commonly-used measures in the literature appears to be particularly challenging for this population. In studies focusing solely on the benefits of dog walking and physical exercise, the detailed and complicated measures may be appropriate. Since this is not the focus of our research, we recommend use of less complex and taxing measures that still offer valid and reliable indicators of physical activity.

Our study offers tentative evidence suggesting there may be an association between dog ownership and well-being. While not the primary objective of this study, results suggest future research should explore this potential relationship with more rigorous methods and a larger sample to examine the potential well-being benefits from owning a dog for people with chronic pain. In particular, further exploration is warranted to understand if dog ownership might be a mediating factor for depression for people with chronic pain, and the mechanism of this action. The new and adapted measures of the benefits of dog ownership, such as providing pain relief and support, for people with chronic LBP show promising results. While we are cautious in concluding that dog ownership has a direct effect on chronic pain, we recommend further investigation of this relationship. These findings will inform the next stage of this research program that will involve a larger, more representative population survey of people who have chronic low back pain to assess more rigorously the potential benefits of dog ownership.

## Figures and Tables

**Table 1 ijerph-16-01472-t001:** Comparison of Dog Owners (*n* = 20) versus Non-Dog Owners (*n* = 36) by Pain Severity, Physical Functioning and Demographic Characteristics ^a^.

Variables	Dog Owners (*n* = 20)	Non-Dog Owners (*n* = 36)
Numerical Pain Index (NPI) Pain Severity (Range = 0–10)	6.40 (1.67)	7.00 (1.45)
Physical Functioning (Range = 0–100) ^b^	56.95 (11.23)	56.81 (15.42)
● Pain Intensity (Range = 0–5)	3.65 (0.93)	3.74 (0.95)
● Pain Walking (Range = 0–5)	2.65 (0.93)	2.94 (0.92)
Days of Poor Physical Health (Range = 0–31)	19.56 (10.66)	18.81 (11.72)
Sex (% Male)	40%	36%
Age	58%	58%
Education (% with University Degree)	42%	36%
Income (1 = impossible/difficult to manageall the time to 4 = easy to manage)	2.40 (0.88)	2.39 (0.90)
Housing (% House vs. Apartment, Farm, Other)	75%	65%
Urban vs. Rural (% Urban)	79%	80%
Employment Status (% Employed)	20%	17%
Marital Status (% Married)	25%	37%
Parental Status (% with children < 18 at home)	21%	20%
Any Dependents (% Yes)	5%	12%

^a^ Note that there are no significant differences between the two groups at the 0.05 level; ^b^ A score of 41%–60% on the Modified Oswestry Low Back Pain Questionnaire indicates “severe disability” where daily living activities are affected by pain.

**Table 2 ijerph-16-01472-t002:** Original and Adapted Measures Developed for Dog Owners (*N* = 20).

Scale (Cronbach’s α)	Items	Response Set	Item Source
**Dog as Pain Reliever** (α = 0.72)	Spending time with my dog reduces my physical pain. My dog seems to know when my pain is at its worst. My dog provides a positive distraction from my pain. My dog takes my mind off my pain. My dog knows when I’m in pain.*	1 = strongly disagree to 5 = strongly agree	Adapted from Marcus et al., (2012) themes verbalized by therapy dog participants (see Table 4) * Original item developed from interviews
**Dog as Stress Reliever** (α = 0.83)	Spending time with my dog helps bring me away from my stress. Petting my dog gets rid of my stress. My dog’s attitude and responses are soothing.*	1 = strongly disagree to 5 = strongly agree	Adapted from Marcus et al., (2012) themes verbalized by therapy dog participants (see Table 4) * Original item developed from interviews
**Dog as Unconditional Support** (α = 0.72)	My dog is not judgemental.* My dogs listens to me.* My dog allows me to cry when I need to.* My dog provides me with unconditional love.* My dog asks for nothing in return. *	1 = strongly disagree to 5 = strongly agree	* Original item developed from interviews
**Dog Gives Meaning and Purpose** (α = 0.95)	My dog makes me hopeful about the future. My dog helps me to reach my goals in life. My dog gives my life meaning. My dog gives my life purpose. My dog gives me a reason to keep going.*	1 = never to 5 = most of the time	Adapted from PROMIS Parent Proxy Bank (2015) V1.0 Meaning and Purpose Short Form 4A * Original item developed from interviews
**Dog Provides Structure and Routine** (α = 0.81)	Caring for my dog gives my life structure.* Caring for my dog requires following a certain routine. *	1 = never to 5 = most of the time	* Original item developed from interviews

**Table 3 ijerph-16-01472-t003:** Comparison of Dog Owners (*N* = 20) versus Non-Dog Owners (*N* = 36) and Zero-Order Correlations with Depression.

Variables	Dog Owners (*N* = 20) Mean (SD)	Non-Dog Owners (*N* = 36) Mean (SD)	Correlation with Depression (*N* = 56)
NPI Pain Severity (Range = 0–10)	6.40 (1.67)	7.00 (1.45)	0.34 *
Physical Functioning (Range = 0–100) ^a^	56.95 (11.23)	56.81 (15.42)	0.32 *
Pain Intensity (Range = 0–5)	3.65 (0.93)	3.74 (0.95)	0.26
Pain Walking (Range = 0–5)	2.65 (0.93)	2.94 (0.92)	0.38 *
Days of Poor Physical Health (Range = 0–31)	19.56 (10.66)	18.81 (11.72)	0.29 *
Loneliness (Range = 1–5)	2.81 (1.38)	3.32 (1.51)	0.64 **
Companionship (Range = 1–5)	3.62 (1.15)	2.99 (1.27) *	−0.49 **
Emotional Support(Range = 1–5)	3.64 (0.98)	3.24 (1.26)	−0.59 **
Depression (Range = 1–5)	2.14 (0.79)	2.73 (1.10) **	

* Significant at the 0.05 level; ** significant at the 0.01 level; ^a^ A score of 41%–60% on the Modified Oswestry Low Back Pain Questionnaire indicates “severe disability” where daily living activities are affected by pain.

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
