# Peer review of "Evaluating the Relationship between Well-Being and Living with a Dog for People with Chronic Low Back Pain: A Feasibility Study"

_ijerph, 2019, doi:10.3390/ijerph16081472_

Round 1

Reviewer 1 Report

This is a very well written paper that presents some interesting results. Given that this was a feasibility study, the discussion and conclusions could have been strengthened with more detail about recommendations and moving forward. Perhaps this could have been achieved by spending a little less time on describing materials and methods, and more on discussion and conclusion. I understand the need to include a detailed study description in a feasibility study, but I felt it was at the expense of providing some well founded recommendations, conclusions and limitations.

Please note, lines 74-79 are repeated in lines 85-90. 

Author Response

Thank you - we have responded to your helpful comments in the file attached.

Reviewer 2 Report

I read with much interest this paper even if I think some improvement are needed for publication.

Introduction. Line 65-79 is a repetition of line 80-90. The dog-human bond description and significance could be improved and enriched with more references.

Materials and Methods. I would like a re-style of this section to improve readability. The authors can use some subtitle to divide it. They can identify a sub-section for ethical statement, a sub-section to describe the sample, a sub-section to describe the development of the questionnaire, a sub-section to describe the study design, which is not clear in the text, a sub-section about patient engagement, data collection and statistical analysis.

The methodology used to choose the measures is not well explained (e.g. it lacks of references).

I can’t find Appendix A 

Results: I don’t find in the sample description information about the results of each scales used in the survey.

Completion Rate should be better analyze reporting results for each survey section.

Discussion and conclusion: they may be improved. This is a feasibility study so I suppose that all point that didn’t work properly in the survey should be the object of critical analysis by the authors.

Author Response

    Thank you for your helpful feedback and attach our responses in a Word file.

Reviewer 3 Report

Very interesting article that contributes to the area of pet ownership and healthy status. It emphasizes that survey data is challenging to collect. Caution should be taken to overinterpret findings that dog ownership has any causation or direct effect on chronic pain, but warrants further investigation. There appears to be a minefield of data to analyze, most of which were not presented in the results. The methods thoroughly discuss how the instruments were chosen and utilized. Bias could be related to how the survey was presented to participants – if it was asking specifically to “investigate the effects of pet ownership on low back pain,” you would likely have biased response rates and responses.  

Abstract

26 – may “be associated with” instead of may “have” 

Introduction

76– what is defined as “patient burden” – can this really be characterized through this study?

80-90 – repeated, but slightly different between the two paragraphs. Lines 80-90 appear more complete than 65-79.

Methods

108 – describe this pain “program”

118-119 – please cite these measures from the literature 

197 – Describe how you presented the study to participants. In what context was the survey presented - Were they told this was about dog ownership? Were they told the survey would take approximately ______ amount of time to complete? Incentive? Were they sent reminders or called?

Results 

234 – how many respondents were from each of the stages of chronic LBP in their “pain programs” discussed in methods?

249-253 – feels like comments deserve to be discussed separately from “recruitment and response rate” and be descriptive – perhaps its own 3.6 section at the end of the methods and described more in-depth. 

254: Please provide a table comparing the demographic information between 2 groups

262 – which items were most frequently not completed?

Table 1 - were all items asked in that order with the headings? Could lead to bias

285 – Recommend that significance at 0.1 not be discussed. This “trending towards significance” misleads readers and leads to overstatements about results. Simply put, if it is >0.05, there was no significance. 

286 – perhaps the bias of this study was that participants who took this survey wanted to participate because they experienced higher levels of pain and had an outlet through this survey to convey this experience.

Table 2: is there supposed to be a significance at 0.01 level?

Recommend removing significance at 0.1 level (or if you do, provide the p value for each value)

Discussion:

Appreciate the brevity of the discussion. At the same time, be sure to thoroughly address each of your objectives.

Objective 1 – was the recruitment strategy effective? Please elaborate if you would change strategy (ie., incentivize, shorten, send reminders, etc.)

Objective 2 – paper questionnaire was appropriate – this is difficult to answer if you did not do a direct comparison between paper vs electronic vs in-person interview. You had a wide range of age of respondents – perhaps comment on challenges (or advantages) that elderly may have in completing surveys

Objective 3 – key variables were valid and reliable for this population – addressed adequately in the discussion

Secondary objective – explore whether a relationship exists between dog ownership and well-being 

This is a very complicated objective to address, especially since so much information is not presented (ie., breed of dog, attachment (LAPS was collected but not discussed), nature of engagement, length of time owned, extent of primary care for the dog, any exercise done with the dog, other animals in the house) and well-being is multifactorial (especially in this context, knowing about current medications and concurrent therapies).  It may be acceptable to state that preliminary evidence shows that there may be an association…but certainly not causation – and that was not the specific objective, nor one that can be appropriately answered with individual surveys.  

319 – It may be bold to state that lower response rate is due to high level of pain and disability. It would have been helpful to ask non-respondents why they did not return surveys or compare sending out surveys to those with high pain vs low pain. 

325 – was this because the survey was framed in a way that was asking to compare dog ownership vs not? Therefore, dog owners more compelled to say something positive about their dogs. 

Please elaborate on the open ended comments, especially about people who have owned dogs in the past and barriers to owning one currently. You may find a difference between those that currently own a dog and ones that do not own a dog but want one and ones that do not own a dog and have no interest in owning one. 

384 – may “be associated with” instead of may “have”

Author Response

Thank you for your helpful feedback. We have attached a Word file with our responses.
